# Correlation between Adverse Events and Antibody Titers among Healthcare Workers Vaccinated with BNT162b2 mRNA COVID-19 Vaccine

**DOI:** 10.3390/vaccines10081220

**Published:** 2022-07-30

**Authors:** Itzchak Levy, Einav Gal Levin, Liraz Olmer, Gili Regev-Yochay, Nancy Agmon-Levin, Anat Wieder-Finesod, Victoria Indenbaum, Karin Herzog, Ram Doolman, Keren Asraf, Rebecca Halperin, Yaniv Lustig, Galia Rahav

**Affiliations:** 1The Infectious Diseases Unit, Sheba Medical Center, Tel-Hashomer, Ramat Gan 52621, Israel; anat.wieder@sheba.health.gov.il (A.W.-F.); rebecca.halperin@sheba.health.gov.il (R.H.); galia.rahav@sheba.health.gov.il (G.R.); 2Sackler Faculty of Medicine, Tel-Aviv University, Tel Aviv 69978, Israel; einav59@gmail.com (E.G.L.); gili.regev@sheba.health.gov.il (G.R.-Y.); nancy.agmonlevin@sheba.health.gov.il (N.A.-L.); yaniv.lustig@sheba.health.gov.il (Y.L.); 3Infection Prevention & Control Unit, Sheba Medical Center, Tel Hashomer, Ramat Gan 52621, Israel; 4Bio-statistical and Bio-mathematical Unit, The Gertner Institute of Epidemiology and Health Policy Research, Sheba Medical Center, Tel-Hashomer, Ramat Gan 52621, Israel; lirazo@gertner.health.gov.il; 5Clinical Immunology, Angioedema and Allergy Unit, The Zabludowicz Center for Autoimmune Diseases, Sheba Medical Center, Tel Hashomer, Ramat Gan 52621, Israel; 6Central Virology Laboratory, Ministry of Health and Sheba Medical Center, Tel-Hashomer, Ramat Gan 52621, Israel; viki.indenbaum@sheba.health.gov.il; 7ARC Innovation Center, Tel Hashomer, Ramat Gan 52621, Israel; karin.herzog@sheba.health.gov.il; 8The Dworman Automated-Mega Laboratory, Sheba Medical Center, Tel-Hashomer, Ramat-Gan 52621, Israel; ram.doolman@sheba.health.gov.il (R.D.); keren.asraf@sheba.health.gov.il (K.A.)

**Keywords:** BNT162b2, immunogenicity, reactogenicity, COVID-19, vaccination

## Abstract

**Objectives:** The BNT162b2 mRNA COVID-19 vaccine has been found to be highly effective in preventing COVID-19 but is associated with increased reactogenicity. We aimed to examine the correlation between immunogenicity and reactogenicity of the BNT162b2 vaccine. **Methods:** Subjects without prior SARS-CoV-2 infection that participated in active surveillance after being vaccinated with the BNT162b2 vaccine were included. Study participants reported adverse drug reactions (ADRs) through questionnaires administered by text message after receiving each dose of the vaccine. A reactogenicity score was developed based on the type and duration of ADRs. In addition, anti-receptor binding domain (RBD) levels and neutralization assays were performed 7–21 and 7–38 days after the first and second vaccine doses, respectively. Associations between ADRs and antibody levels were assessed by Spearman correlations. Multivariable logistic regression analyses were used to identify factors associated with ADRs. **Results:** A total of 831 health care workers were included. The mean age was 46.5 years (SD = 11.8) and 75.5% were females. 83.4% and 83.3% had at least one local ADR after the first and second doses, respectively. 33% and 83.2% had at least one systemic ADR after the first and second doses, respectively. Multivariate logistic regression analysis found a significant correlation between ADR score and anti-RBD-IgG titers (r = 0.366; *p* < 0.0001) after adjustment for age, gender, and days after the second vaccination. High anti-RBD-IgG levels, being younger than 55 and being female, were all correlated with increased rates of ADRs. **Conclusion:** BNT162b2 mRNA COVID-19 vaccine reactogenicity appears to be correlated with higher post-vaccination antibody levels and is independently associated with younger age and female gender.

## 1. Introduction

More than 48% of the world population has been vaccinated against COVID-19 as of 22 December 2021, totaling more than 8.8 billion doses. The vaccination rollout with the BNT162b2 mRNA vaccine was initiated in Israel on 19 December 2020 and by 7 September 2021, 60.9% of the eligible Israeli population had received two doses.

In the phase 3 clinical trial on the safety and efficacy of the BNT162b2 vaccine, adverse drug reactions data were collected by registration through an electronic diary (solicited) or by participant report without prompts using an electronic diary (unsolicited). It was found that reactogenicity was common, but mostly mild [1]. ADRs included local as well as systemic reactions, such as fatigue, headache, and fever. Systemic ADRs were more common after the second dose of the vaccine. Severe ADRs were rare.

In comparison to clinical trials, real-world data shows variable rates of ADRs after vaccination with the BNT162b2 mRNA COVID-19 vaccine. In a British prospective observational study, 71.9% and 68.5% of vaccinated people developed local adverse events after the first and second dose, respectively, and 13.5% and 22% developed systemic adverse events after the first and the second dose, respectively [2]. Furthermore, prior infection with SARS-CoV-2 was associated with higher rates of local and systemic ADRs. In a study conducted on 3078 health care workers (HCW) that were vaccinated with the BNT162b2 vaccine, 59.6% and 73.4% developed any ADRs (local and/or systemic) after the first and second dose, respectively. Prior infection with SARS-CoV-2 before vaccination increased the risk of moderate or severe ADRs by at least three-fold [3].

The BNT162b2 vaccine study, conducted on over 2 million people in Israel, found that the vaccine is safe and causes significantly fewer serious ADRs than COVID-19 [4].

Associations between ADRs and immunogenicity of the vaccine have been assessed in several studies. An association between systemic ADRs and higher anti-S protein was found in two Japanese studies that examined factors associated with adverse events among healthcare workers. In the first study [5], a significant positive correlation between higher body temperature and higher anti-SARS-CoV-2 antibody titer was observed 3 months after vaccination but not at 6 months. In the second study [6] it was found that systemic adverse events, specifically muscle and joint pain after the second dose, were associated with elevated anti-S1 protein IgG antibody titer and neutralizing activity.

However, in other studies, only a weak correlation or no correlation at all between immunogenicity and reactogenicity was found [7,8,9].

Given the mixed results of data on this topic, this study aims to evaluate ADRs and antibody titers following vaccination with the BNT162b2 vaccine and determine the correlation between immunogenicity and reactogenicity.

## 2. Methods

### 2.1. Study Design and Population

This is a prospective observational study conducted from 21 December 2020 to 15 April 2021 at Sheba Medical Center (SMC) as part of an active surveillance of SMC health care workers (HCWs) vaccinated with the BNT162b2 mRNA vaccine. This study was approved by the Sheba Ethics Committee (SMC 8008-20).

SMC is the largest tertiary medical center in Israel, with 1600 beds and 14,479 HCW, all of which were invited to participate in this study. Vaccinated HCWs were included in the study if they provided written informed consent, completed the digital ADR form, had at least one serological test after the first dose of the vaccine, and had a negative anti-SARS-CoV-2 IgG assay before receiving the first dose. HCWs with a positive SARS-CoV-2 PCR test before vaccination or a confirmed infection with COVID-19 following vaccination were excluded from the study. Immunocompromised HCW and those with an autoimmune disease were also excluded.

Serological tests were performed before administration of the first vaccine dose, 1–2 weeks after the first dose, at week 3 with the administration of the second dose, and 1–2 weeks after the second dose. The time interval between the first and second vaccine doses was 21 days. Seven days after each vaccine dose, study participants received a text message survey on their personal cell phones that queried their response to the vaccine. Solicited ADRs collected included localized and systemic side effects, the duration of these symptoms, and whether the participant required medical care or hospitalization. Unsolicited serious and non-serious adverse events were collected up to a month after the second dose. In addition, demographic data such as age, sex, and underlying comorbidities were collected.

### 2.2. Reactogenicity

A reactogenicity score was developed to rate the number and duration of systemic adverse events participants had to the vaccine (Table 1), with a higher score indicating more adverse events. Systemic adverse events were selected based on the most common reactions that presented in the phase 3 study of the vaccine [1]. Each event received one or two points according to the assumed association of the reaction with the vaccine. For example, fatigue, which is less typical and may be attributed to other factors, received 1 point for each day reported, whereas fever and myalgia, which are more typically associated with reactogenicity, received 2 points for each day they were reported. Local reactions were not included in the total score as they are usually attributed to mechanical factors (e.g., local damage to the muscle or reaction to ancillary materials such as preservatives, stabilizers, or adjuvants).

### 2.3. Immunogenicity

Blood samples were tested using the SARS-CoV-2 anti-RBD IgG assay (Beckman Coulter, CA, USA). In addition, a SARS-CoV-2 pseudo-virus (psSARS-2) neutralization assay was performed as described [10] to detect SARS-CoV-2 neutralizing antibodies (NA) using a green fluorescent protein (GFP) reporter-based pseudotyped virus with a vesicular stomatitis virus (VSV) backbone coated with the SARS-CoV-2 spike (S) protein [generously provided by Dr. Gert Zimmer (Institute of Virology and Immunology (IVI), Mittelhäusern, Switzerland]. Sera not capable of reducing viral replication by 50% at a 1 to 8 dilution or below were considered non-neutralizing.

### 2.4. Statistical Methods

Data analysis was performed using SAS 9.4 software (Cary, NC, USA). Descriptive statistics were expressed as percentages for categorical data or mean ± standard deviation (SD) and median (interquartile range) for normally or non-normally distributed continuous data, respectively. Anti-RBP-IgG titers were expressed as geometrical mean titers (GMT). Since none of the variables were normally distributed, Spearman correlation coefficients were used to measure degrees of association between ADR scores as well as individual ADRs and anti-RBP-IgG titers. McNemar’s test and the Wilcoxon Signed Rank test were used to compare statistical differences in the ADRs proportion or ADR score medians between the two time points (after 1st and 2nd vaccinations).

Multivariable logistic regression analysis was used to identify factors associated with ADRs. The variables included gender, age, and RBD-IgG titers. Results were presented as odds ratios (OR), 95% confidence intervals (CI), and *p*-values. A *p*-value less than 0.05 was considered statistically significant.

## 3. Results

### 3.1. Participants

Between 20 December 2020 and 15 April 2021, 91% (13,212/14,519) of SMC personnel received two doses of the BNT162b2 vaccine. Of these, 12,582 received a text message asking them to participate in this study. 2240 HCWs responded to the questionnaire of which 124 were excluded due to immunosuppression or autoimmune diseases and 37 were excluded due to incomplete data. There was a higher response rate among women compared to men both after first (18.7% vs. 15.5%, *p* < 0.0001) and second (14.2% vs. 9.9%, *p* < 0.0001) doses.

738 HCWs completed the ADR questionnaires and had serological tests after the first or second vaccine, respectively, and were included in the final analysis.

The mean age of study participants was 46.5 ± 11.8 years (range: 20–82.3) and 75.5% (*n* = 627) were females. 18.5% (*n* = 154) were physicians, 22.3% (*n* = 185) nurses, 35% (*n* = 291) paramedical, and 24.2% (*n* = 201) administrative workers. Blood for anti-RBD-IgG antibodies was drawn 11 ± 3.6 days and 13.7 ± 5.6 days after the first and the second doses, respectively. Blood for NAs was drawn from 71 participants 14.1 ± 0.4 days after the first dose and from 118 participants 6.9 ± 0.6 days after the second dose.

### 3.2. Adverse Drug Reactions

In total, 85.6% and 91.2% had any adverse event after the first and second doses, respectively. Yet, these were mostly local reactions. There were no significant differences in the rates of local ADRs following the first and second vaccines (83.4% and 83.3%, respectively). However, systemic ADRs were significantly more frequent after the second dose (83.2% vs. 33%r): axillary lymphadenopathy (6.5% vs. 1%, *p* < 0.0001), fever > 38 °C (7.2% vs. 0.6%, *p* < 0.0001), fatigue (48.6% vs. 20.9%, *p* < 0.0001), myalgia (37.1% vs. 11.4%, *p* < 0.0001), headache (37.8% vs. 15%, *p* < 0.0001), arthralgia (14.2% vs. 2.8%, *p* < 0.0001), and non-facial paresthesia (3.1% vs. 1.1%, *p* = 0.001). The median (Q1–Q3) score reflecting all systemic ADRs and their duration was significantly higher after the second dose (2.5 (0–6)) than the first dose (0 (0–2)). Antipyretics were used significantly more after the second dose (30.1% vs. 10.3%, *p* < 0.0001). Following the second vaccine dose, HCWs missed significantly more workdays than following the first dose (7.2% vs. 0.2%, *p* < 0.0001 for missing 1 day and 3.5% vs. 0.2%, *p* < 0.0001 for missing 2 or more days), (Table 2, Figure 1).

Pain at the injection site was significantly more common in HCWs younger than 55 years versus those older than 55 years (89.8% vs. 64.9%, *p* < 0.0001 after the first and 87.8% vs. 69.1%, *p* < 0.0001 after the second dose). Headache and myalgia were more common among HCWs younger than 55 years old (42.5% vs. 23.4%, *p* < 0.0001 and 41% vs. 24.6%, *p* < 0.0001, respectively), (Appendix A).

After the second vaccine dose, ADRs including pain and warmth at the injection site, axillary lymphadenopathy, fatigue, headache, myalgia, arthralgia, and the need for antipyretics or analgesics were significantly more common in females (Appendix A).

### 3.3. Immunogenicity

Anti-RBD IgG titer increased significantly from a GMT of 0.11 S/CO (95% CI: 0.08–0.14) 11.7 ± 3.6 days after the first dose to a GMT of 32.55 S/CO (95% CI: 31.11–34.05) 13.7 ± 5.6 days after the second vaccination (Table 3). Anti-RBD IgG GMT after the second dose was significantly higher among females versus males: GMT of 33.5 S/CO (95% CI: 33.85–37.2) vs. 26.9 S/CO (95% CI: 24.1–30.1) (*p* < 0.0001). IgG anti-RBD GMT after the second dose was significantly higher among HCWs younger than 55 years versus older HCWs: 35.62 S/CO (95% CI: 34.1–37.2) vs. 26.62 S/CO (95% CI: 23.44–30.2) (*p* < 0.0001).

Neutralizing antibodies GMT increased from 23.41 (95% CI: 17.95–30.54) 14.1 ± 0.4 days after the first dose to a GMT of 745 (95% CI: 611.5–908.9) 6.9 ± 0.6 days after the second vaccination (Table 3). Neutralizing titer after the second dose was significantly higher among females compared to males: 923.6 (95% CI: 734–1161) vs. 362 (95% CI: 153–858), respectively, but did not differ between younger and older HCWs. However, NA was performed in only 54 and 17 HCWs under 55 and over 55 years old, respectively, after the first vaccine and in 78 and 40 HCWs under 55 and over 55 years respectively after the second vaccine.

### 3.4. Correlation between Adverse Effects, Anti-RBD IgG, and Neutralizing Antibodies GMT

Spearman correlation, adjusted for age, gender, and days after second vaccination, found a significant correlation between systemic ADR score and anti-RBD-IgG titers (*R* = 0.366; *p* < 0.0001) and a weaker correlation with neutralization antibodies (*R* = 0.283; *p* = 0.005). There was no correlation between ADRs and immunogenicity after the first vaccine dose (Table 4, Figure 2).

### 3.5. Factors Associated with Adverse Events following Vaccination Using Multivariate Logistic Regression Analysis

Multivariate logistic regression analysis found that following the first vaccine dose, being female and under 55 years of age were associated with increased risk of any ADRs (OR = 2.38, 95% CI: 1.01–5.55; OR = 3.29, 95% CI:1.47–7.39, respectively), whereas IgG anti-RBD titers did not influence the risk of ADRs. However, after the second vaccine dose, being a female, younger than 55 years old and having an increased IgG anti-RBD GMT were significantly and independently associated with increased risk of any adverse events (OR = 2.86, 95% CI: 1.6–5.1, *p* = 0.0004; OR = 3.18, 95% CI: 1.83–5.52, *p* < 0.0001; OR = 1.36, 95% CI: 1.33–1.39, *p* = 0.0029, respectively), (Table 5).

## 4. Discussion

This study found a significant correlation between reactogenicity and immunogenicity after adjusting for age and sex among 831 health care workers vaccinated with the BNT162b2 mRNA vaccine. Systemic ADRs were more common after the second dose of the vaccine. Gender and age independently affected antibody levels and the magnitude of ADRs.

Several studies have looked at correlations between reactogenicity and immunogenicity among recipients of mRNA anti-COVID-19 vaccine and mixed results were found. Our study found a weak but significant association. In two separate studies, Koike and Kobashi in Japan did find a significant correlation of immunogenicity reflected by anti-SARS-CoV-2 S protein IgG antibodies titer and reactogenicity after the second dose of the vaccine. In these studies, the correlation was found to be only with some of the ADRs. In the first study [5], a significant positive correlation was found between higher body temperature and higher antibody titer 3 months but not 6 months after vaccination, and in the second study [6], a significant correlation was found between muscle and joint pain and anti-S1 protein IgG antibody titer and neutralizing activity.

However, in another study Zhang et al. [7] looked for a correlation between neutralizing activity against SARS-CoV-2 after vaccination with BNT162b2 or Coronavac vaccine, which is a whole inactivated virus COVID-19 vaccine, and found only a low correlation between AEs and the BNT162b2 vaccine.

Takeuchi et al. found no correlation between reactogenicity and antibody production in a study of 67 HCWs [8], while Held et al. found that adverse events were weakly correlated with spike protein antibody levels after vaccination with BNT162b2 vaccine in a study of 80 HCWs [9]. Hwang et al. did not find an association between local or systemic reactogenicity and humoral immunogenicity in individuals who received either the BNT162b2 mRNA or the AZD1222 vaccine [11]. However, a weak correlation was found between reactogenicity and immunogenicity following the herpes zoster vaccine [12].

Several studies have demonstrated that individuals who recovered from COVID-19 had increased reactogenicity following vaccination and higher titers of RBD-IgG compared to those who were vaccinated and not infected [2,3], or were infected but had mild disease [13]. As expected, anti-S-RBD IgG as well as neutralizing antibodies titers increased after each vaccination dose. Local ADRs usually result from local resident immune cell activation (such as macrophages, dendritic, and mast cells) by the adjuvant or lipid nanoparticle in mRNA vaccines used to stabilize the mRNA. Systemic reactogenicity results from spillage of inflammatory mediators or products into the circulation or via immune system activation by the protein used as the antigen (e.g., S protein) [13]. The latter may be more profound after provocation with the second dose. This may explain why we found a correlation between immunogenicity and reactogenicity with only systemic ADRs after the second dose, but not the first dose.

Our study found a higher rate of ADRs compared to the rate reported in the drug registration study. Other studies have also reported higher rates [14]. Different modes of reporting ADRs or selection bias could explain these findings as people who experience ADRs are more likely to participate in solicited questionnaires than those with mild or no ADRs.

Similar to other mRNA vaccine studies, ADRs [1,15,16,17] and reactogenicity [18] were less common among older participants, while younger subjects had higher levels of anti-RBD IgG levels after the second dose [19,20]. Immunosenescence and aging may explain these findings [21]. Immunosenescence decreases the ability of both CD4+ and CD8+ cells to function correctly, lowers naïve T cell frequency, expands memory T cells, and shrinks T cell diversity [22]. Aging changes the microenvironment and regulation of developmental checkpoints, resulting in quantitative and qualitative changes in B cell generation [23], as well as impaired replenishment of peripheral B cells, reduction in regenerative B cell capacity and, ultimately, humoral responses. Decreased vaccine effectiveness in the elderly has been demonstrated after vaccination against influenza [24] and PPV23 [25].

Our study also found that females reported more ADRs than males even after adjusting for age and professional sector. This may be related to higher responses to questionnaires among women than men; indeed, we found a higher response rate to text messages in women. Both registration studies of mRNA vaccines and real-life studies showed a higher rate of ADRs among females. Higher reactogenicity in women was also shown in other vaccination studies such as influenza and diphtheria, tetanus and pertussis (DTP) [26,27].

### Limitations

Selection bias was a major limitation since only HCWs with high compliance participated in the study, and those responding to the questionnaires may have had increased ADRs. Nevertheless, this should not influence the correlation between reactogenicity and immunogenicity.

In summary, we found that vaccination with the BNT162b2 mRNA vaccine induces expected local and systemic ADRs. In addition, we found independent correlations between reactogenicity and younger age, female sex, and antibody levels. This result does not point to causality, and the mechanism of this association is yet to be shown. Further studies are needed to further understand whether there is causality in either direction or if and what is the indirect association.

## Figures and Tables

**Figure 1 vaccines-10-01220-f001:**
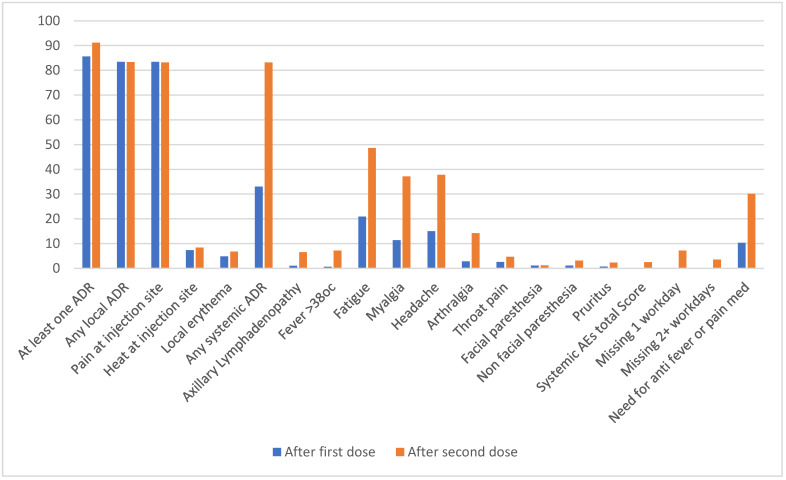
Adverse events rate after first and second vaccination (%).

**Figure 2 vaccines-10-01220-f002:**
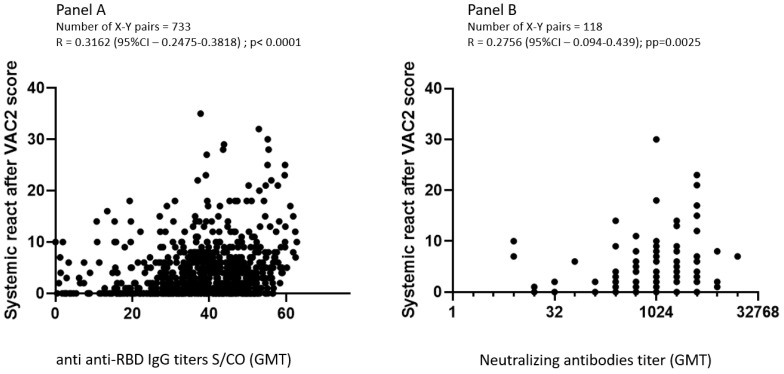
Correlation between reactogenicity score and immunogenicity as measured with anti-RBD IgG (**panel A**) and neutralizing antibodies (**panel B**), Spearman correlation coefficient.

**Table 1 vaccines-10-01220-t001:** Adverse events score system.

Adverse Event	Number of Points for Each Day of the Event
Fatigue	1
Headache	1
Myalgia	2
Fever > 38 °C	2
Arthralgia	1
Local Lymphadenopathy	1
Systemic rash	1
Pruritus	1
Facial paresthesia	1
Non-facial paresthesia	1
Need for antipyretic or analgesic medication	2
Total score	Sum of the above

**Table 2 vaccines-10-01220-t002:** Adverse events rate after first and second vaccination.

Adverse Event	After 1st Dose *N* = 831 (100%)	After 2nd Dose *N* = 738 (100%)	*p* *
At least one adverse event	711 (85.6)	673 (91.2)	0.0003
Any local AE	693 (83.4)	615 (83.3)	0.5807
Pain at injection site	693 (83.4)	614 (83.2)	0.5279
Heat at injection site	61 (7.3)	62 (8.4)	0.4855
Local erythema	40 (4.8)	50 (6.8)	0.1229
Any systemic AE	274 (33.0)	614 (83.2)	<0.0001
Axillary Lymphadenopathy	8 (1.0)	48 (6.5)	<0.0001
Fever > 38 °C	5 (0.6)	53 (7.2)	<0.0001
Fatigue	174 (20.9)	359 (48.6)	<0.0001
Myalgia	95 (11.4)	274 (37.1)	<0.0001
Headache	125 (15.0)	279 (37.8)	<0.0001
Arthralgia	23 (2.8)	105 (14.2)	<0.0001
Throat pain	22 (2.6)	35 (4.7)	0.0159
Facial paresthesia	9 (1.1)	9 (1.2)	0.7963
Non-facial paresthesia	9 (1.1)	23 (3.1)	0.0025
Pruritus	6 (0.7)	17 (2.3)	0.0105
Systemic AEs total Score—median (Q1–Q3)	0 (0–2)	2.5 (0–6)	<0.0001
Missing 1 workday	2 (0.2)	53 (7.2)	<0.0001
Missing 2 + workdays	2 (0.2)	26 (3.5)	<0.0001
Need for anti-fever or anti-pain medication	86 (10.3)	222 (30.1)	<0.0001

* By McNemar’s test.

**Table 3 vaccines-10-01220-t003:** Anti-RBD IgG and neutralizing antibodies titers following first and second vaccine. doses.

Assay	After 1st Vaccine *	After 2nd Vaccine **
Days for Anti-RBD–IgG; mean ± SD (range)	11.7 ± 3.6 (7–21)	13.7 ± 5.6 days (7–38)
Anti-RBD-IgG (Geometric mean (CI95%))	0.11 (0.08–0.14)	32.55 (31.11–34.05)
Days for neutralizing Ab; Mean ± SD (range)	14.1 ± 0.4 (13–17)	6.9 ± 0.6 (4–10)
Neutralizing Ab (Geometric mean (CI95%))	23.41 (17.95–30.54)	745 (611.5–908.9)

* *N* = 227 participants for anti-RBD-IgG and 71 participants for neutralizing Ab. ** *N* = 823 participants for anti-RBD-IgG and 185 participants for neutralizing Ab.

**Table 4 vaccines-10-01220-t004:** Spearman Correlation between adverse events and immunogenicity after the first and the second vaccines adjusted for age, gender, and days after vaccination.

	Anti-IgG-RBD				Neutralizing AB			
	After 1st Vaccination		After 2nd Vaccination		After 1st Vaccination		After 2nd Vaccination	
*N*	233		733		89		118	
	*R*	*p*	*R*	*p*	*R*	*p*	*R*	*p*
Score of systemic AEs	0.1	0.13	0.366	<0.0001	−0.1	0.34	0.283	0.005
Axillary lymphadenopathy	−0.018	0.78	0.15	<0.0001	NA		0.1	0.28
Fever	0.12	0.7	0.19	<0.0001	−0.2	0.057	0.13	0.15
Fatigue	0.1	0.12	0.18	<0.0001	−0.08	0.46	0.18	0.04
Headache	0.07	0.04	0.19	<0.0001	0.08	0.45	0.12	0.18
Myalgia	0.14	0.3	0.21	<0.0001	0.017	0.87	0.2	0.03
Arthralgia	0.09	0.15	0.135	<0.05	0.003	0.97	0.03	0.7
Need for antipyretic or analgetic medication	0.078	0.24	0.197	<0.0001	−0.12	0.25	−0.16	0.07
Missing workdays	0.06	0.36	0.11	0.0027	NA		0.06	0.4

**Table 5 vaccines-10-01220-t005:** Factors associated with adverse events following 1st and 2nd vaccination doses using multivariate logistic regression analysis.

	After 1st Vaccination (*N* = 233) C Statistics = 0.719		After 2nd Vaccination (*N* = 733) C Statistics = 0.751	
	OR (CI 95%)	*p*	OR (CI 95%)	*p*
Female	2.38 (1.02–5.55)	0.0441	2.86 (1.6–5.1)	0.0004
Age < 55 years old	3.29 (1.47–7.39)	0.0039	3.18 (1.83–5.52)	<0.0001
IgG-Anti-RBD	1.15 (0.88–1.51)	0.3	1.36 (1.33–1.39)	0.0029
Number of days after second vaccine	0.96 (0.86–1.08)	0.5143	1.01 (0.96–1.06)	0.6905

## Data Availability

The data that support the findings of this study are available from the corresponding author, itsik.levy@sheba.health.gov.il (I.L.) and galia.rahav@sheba.health.gov.il (G.R.), upon reasonable request.

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
