# Peer review of "Correlation between Adverse Events and Antibody Titers among Healthcare Workers Vaccinated with BNT162b2 mRNA COVID-19 Vaccine"

_vaccines, 2022, doi:10.3390/vaccines10081220_

Round 1
Reviewer 1 Report
Reviewer comment
Itzchak et al., investigate the correlation between antibody level and adverse events after two doses of BNT162b2 mRNA vaccination.
comments
1. The introduction of previously published paper related to ADRs are not comprehensive, the authors reported a published British cohort, but in fact, more studies such as
https://pubmed.ncbi.nlm.nih.gov/35335084/, https://pubmed.ncbi.nlm.nih.gov/35687563/,
https://pubmed.ncbi.nlm.nih.gov/35455312/
Please discuss these studies in the introduction and discussion if needed
2. The authors mentioned in the methods section that “a prior IgG test before receiving the first dose which practically rules out a prior infection up to six months” is not true as only the IgN test (nucleocapsid) can rule out prior infection. Please rectify this in the main text
3. The reactogenicity score generated is encouraging; however, the scoring system considers the number of days certain symptoms are shown with corresponding higher scores increased the error rate in the sense that most people who suffer from adverse reactions such as fever will take medicine and some would not. That would bring a bias in which the people who don’t take medicine end up with a higher score. The reviewer would suggest a score is given based on the presence/absence instead of “days”. In addition, “fatigue” can be objective and based on the nature of the daily activities of the recipient as well.
4. The other concern would be the big range of the dates where the blood was collected for IgG testing, as per previous studies, it is established that IgG levels increased exponentially after vaccination, and the time point of when the blood was collected are crucially important.
5. The bias in the number of female participants (>70%) would also cause a bias in the results reporting
6. “Only a few studies have looked at correlations between reactogenicity and immunogenicity among recipients of mRNA anti COVID-19 vaccine and did not find a correlation or only a minimal correlation.” Similar to comment number 1, for example; https://pubmed.ncbi.nlm.nih.gov/35455312/ reported that younger females who produced higher levels of IgG suffer from a more severe form of adverse reaction which agrees with what the authors reported.
Author Response
I would like to thank the reviewer for his important comments.
Here is our response:
1. I added more recent and relevant references to the introduction and omitted non-relevant ones as was suggested by the reviewer. I also discussed these studies in the discussion.
2. Since we did not check for anti-N IgG antibody I deleted the sentence that says that “a prior IgG test before receiving the first dose which practically rules out a prior infection up to six months”.
3. We agree that our "reactogenicity score" may be generated in different ways. The biases that the building of the score may generate were discussed in the discussion. We did not find a reference that will show how to measure objectively fatigue and that is why we decided not to consider it in the score.
4. The reviewer is totally right concerning the big range of the dates where the blood was collected for IgG testing, but since there is nothing we can do with it we discussed it as a limitation of the study.
5. Concerning the bias that is caused by the number of female participants (>70%) we understand the reviewer's concern. we are less concerned because at the final point we did build a model which took this into account.
- We corrected the phrase, widened the discussion, and added the references that were suggested by the reviewer.
Reviewer 2 Report
In their manuscript “
Correlation between adverse events and antibody titers among 2 healthcare workers vaccinated with BNT162b2 mRNA Covid-19 Vaccine”, presented the examination of the correlation between immunogenicity and reactogenicity of the BNT162b2 vaccine. The BNT162b2 vaccine study was conducted over 2 million people in Israel and was found that the vaccine is safe and causes significantly fewer serious ADRs than Covid-19.
After reviewing the manuscript, I found that the manuscript best fit for the readers of the journal significantly and rich in scientific measures. The present work is well-executed; however, the presentation of the manuscript can be improved in several aspects.
a. The corona virus infectivity is changing by rapid mutation, it will be better to include correlation of different corona variant, and formation of antibodies as mentioned in result and discussion section.
b. The data is present in form of table and detail description was carried out in the subsequent paragraph. Like adverse drug reactions (Table 2), where all the findings were mentioned in table and explained in text. It will be better to present the data with the help of figure, thus findings of the authors will be clearer to reader.
c. The ADRs rate found by the authors is higher than rate. It will be better to comment the reasons behind these findings in the abstract section.
d. The authors should make comparison with previously published results. Proper validation of the model is needed also supported by past studies
e. Include some statistical and real data about the relevant work
f. The introduction section should be improved avoiding lump sum references ,all references should be cited with detailed and specific descriptions. Introduction section need to extend it. References should be in uniform numbering , some mathtematical results about covid 19 will improve it
https://doi.org/10.1186/s13662-022-03701-z,2022, https://doi.org/10.1016/j.rinp.2021.104069
This study aims to evaluate ADRs and antibody titers following vaccination with the BNT162b2 vaccine and determine the correlation between immunogenicity and reactogenicity. Such correlation studies are rare in literature and these findings are important for the community to understand the post-vaccination profile.
Based on these finding, the reviewer recommend the article to be publish in Vaccine journal after the amendments suggested above.
Author Response
We would like to thank the reviewer for his/her important comments.
1. Although the coronavirus infectivity is changing by rapid mutation, we can not include the correlation of different corona variants and this was beyond the scope of our study.
2. A figure (Figure 1) which shows more clearly the adverse events rate was drawn as suggested by the reviewer.
3. We added more studies to make a comparison with previously published results as suggested by the reviewer.
4. The difficulty of doing a proper validation of the model is discussed in the discussion.
5. As suggested by the reviewer the introduction section was revised
Reviewer 3 Report
This is a well designed and presented assessment of reactogenity in HCWs' with a previous history of SARS-CoV2 infection following the first and the second dose of BNT162b2 mRNA vaccine. I have no comment.
Author Response
I would like to thank the reviewer for his review.